# Spektr–UF Mission Spectrograph Space Qualified CCD Detector Subsystem

**Andrey Shugarov \*,† and Mikhail Sachkov †**

Institute of Astronomy of the Russian Academy of Sciences, 119017 Moscow, Russia; msachkov@inasan.ru
\* Correspondence: shugarov@inasan.ru
† These authors contributed equally to this work.

**Abstract:** Spektr–UF (World Space Observatory Ultraviolet, WSO-UV) is a Russian-led international collaboration aiming to develop a large space-borne 1.7 m Ritchey–Chretién telescope with science instruments to study the Universe in ultraviolet wavelengths. The WSO-UV spectrograph (WUVS) consists of three channels: two high-resolution channels (R = 50,000) with spectral ranges of 115–176 nm and 174–310 nm, and a low-resolution (R = 1000) channel with a spectral range of 115–305 nm. Each of the three channels has an almost identical custom detector consisting of a CCD inside a vacuum enclosure, and drive electronics. The main challenges of the WUVS detectors are to achieve high quantum efficiency in the FUV-NUV range, to provide low readout noise (3 e− at 50 kHz) and low dark current (<12 e−/pixel/hour), to operate with integral exposures of up to 10 h and to provide good photometric accuracy. A custom vacuum enclosure and three variants of a custom CCD272-64 sensor with different UV AR coatings optimised for each WUVS channel were designed. The enclosure prevents contamination and maintains the CCD at the operating temperature of −100 °C, while the temperature of the WUVS optical bench is +20 °C. A camera electronics box (CEB) that houses the CCD drive electronics was developed. Digital correlated double sampling technology allows for extremely low readout noise and flexible frequency for normal and binned pixel readout modes. This paper presents the WUVS detector design drivers, methods for extending the service life of the CCD sensors working with low signals in a space radiation environment and the key calculated parameters and results of the engineering qualification model qualification campaign.

**Keywords:** ultraviolet; WSO–UV; CCD; quantum efficiency; gradient anti-reflection coating

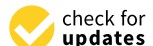



## 1. Introduction

The Spektr–UF (Spectrum–UV, World Space Observatory–Ultraviolet, WSO–UV) observatory is designed to study the Universe in 115 nm to 310 nm ultraviolet (UV) wavelengths. A 170 cm aperture space telescope has two main science instruments—spectrographs and field cameras [1–3].

The main scientific objectives of WSO-UV are: evolution of the early Universe, star formation, galaxy evolution, extrasolar planets and their atmospheres, stellar physics, compact bodies, astrophysical accretion processes, etc. [4].

WUVS (WSO-UV Spectrograph) consists of two high-resolution spectrographs (R = 50,000) covering the far-UV range of 115–176 nm (Vacuum UltraViolet Echellé Spectrograph, VUVES) and the near-UV range of 174–310 nm (UltraViolet Echellé Spectrograph, UVES). The third WUVS channel is a long-slit spectrograph (LSS) that has two sub-channels operated with a single detector, the total coverage wavelength range being 115–305 nm (Figure 1). WUVS high-resolution FUV and NUV channels have a classical Echellé optical design (Figure 2), and the low-resolution channel has a Rowland optical design (Figure 3) with two subchannels for FUV and NUV, both operated with one detector. The VUVES and UVES channel optical schemes are optimised to utilize the whole photosensitive surface

of the CCD detectors [5]. The LSS channel has two long narrow spectra located along the bottom long side of the detector while the remaining part of the CCD remains unusable [6].

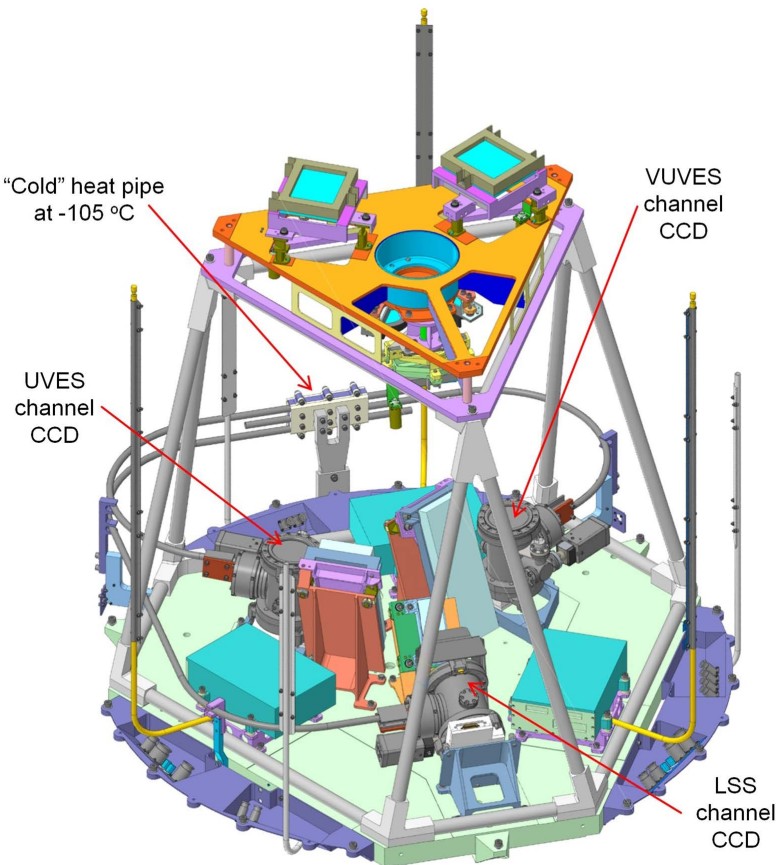

**Figure 1.** WUVS spectrograph with three CCD detectors.

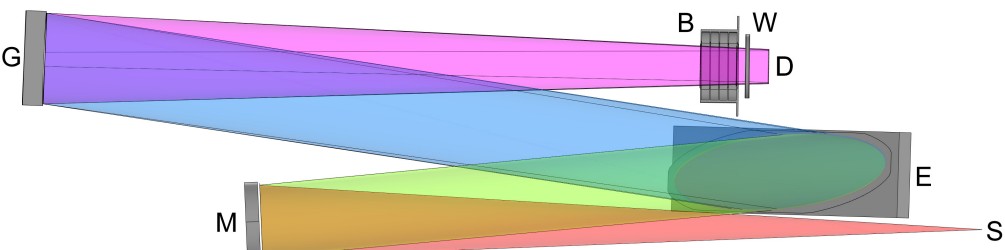

**Figure 2.** UVES optical layout: S—entrance slit, M—collimator mirror, E—echellé grating, G—cross-dispersion grating, W—entrance window of CCD, D—CCD surface, B—detector baffle.

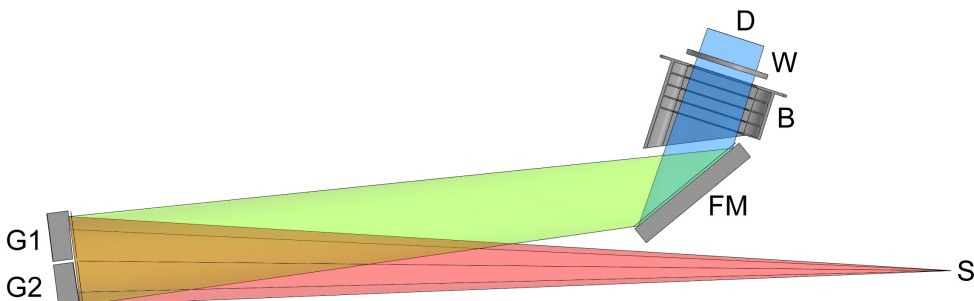

**Figure 3.** LSS Layout: S—entrance slit, G1—toroidal grating of NUV branch, G2—toroidal grating of FUV branch, FM—flat mirror, W—entrance window of CCD, D—CCD surface, B—detector baffle.

The spectrographs' slits have a size of 1 arcsec, while the telescope PSF is three times smaller; therefore, all spectrographs operate in a slit-less mode.

WSO-UV will operate on geosynchronous orbit. One of the advantages of the WUVS spectrograph over similar HST spectrographs operating on the low Earth orbit is the possibility of having long uninterrupted observations of up to 10 h or even longer.

Each WUVS channel has a mechanical shutter close to the slit to block the light from the telescope to allow for CCD readout, which could be as long as 2 min at the lowest CCD readout speed. Another goal of the shutter is to obtain accurate CCD dark current calibration images, which may take up to several hours.

The WUVS does not have an internal lamp for wavelength calibration; therefore, well-known stars are the only source used to calibrate the spectrograph.

Each long WUVS exposure should be split into a series of short sub-exposures to implement cosmic ray rejection algorithms and dithering (if needed) and to compensate for possible minor spectra displacement on the CCD due to the spectrograph thermal stability and spacecraft pointing system drift. A sub-exposure should be no longer than 10–20 min while the total observation time may reach 10 h.

T-170M telescope fine guidance sensors (FGSs) were mounted close to the spectrographs' entrance slits to ensure the proper pointing and stabilisation of the telescope in order to keep the star at the center of the slit. For each 10 min subexposure, the FGS provides information about the actual position of the star within the slit. In case of unexpected problems with the telescope pointing stability during WUVS long exposure (up to 10 h), this information will be used to prevent degradation of the spectral resolution.

To cool down detectors, the WSO–UV spacecraft provides an isolated cold heat pipe connected to a large external radiator. This system maintains the temperature of the CCD enclosure's cold finger at around −105 °C. Low-power heaters are installed on the cold finger and near the CCD for precise temperature stabilisation.

The WUVS data processing unit uses lossless compression algorithms while co-adding images will be a part of the science data processing pipeline at the ground scientific center.

With this article, we want to give visibility to the current progress of developing and qualifying world-class space-qualified custom UV detectors for a large WSO-UV space mission.

The main aim is to present general scientific and technological drivers and key design solutions. Special attention was paid to the problem of CCD operation with very low signals in a space radiation environment and methods for extending the CCD service life. At the end of the paper, key measured parameters of the engineering qualification model are presented.

We hope that our experience in UV technologies will help astronomers to develop other new UV instruments.

## 2. WUVS Detector System General Design

The WUVS detectors should provide high sensitivity in a UV, high dynamic range and low dark current.

The previous design of the WSO-UV spectrograph was based on micro channel plate (MCP) detectors. At that time, it was the only way to achieve high sensitivity in the UV range [7]. The disadvantages of the MCP detectors are limited local and global count rates, lifespan and resolution. MCP detectors limit the maximum signal-to-noise ratio (SNR) of the spectra by 30 approximately. To eliminate the lifespan problem, it was suggested to install a spare detector in each channel as well as a corresponding remotely controlled folding mirror to activate the spare detector.

Thanks to the improvements in CCD technology, especially in UV coatings and readout noise, replacing an MCP detector by a CCD is made possible.

Before choosing the CCD technology to be used in the WUVS instead of MCP, the use of a custom design scientific complementary metal–oxide–semiconductor (sCMOS) was briefly considered. The main possible advantages of the sCMOS are a potentially

lower readout noise (down to 1 e⁻ RMS) and the absence of charge transfer degradation due to the radiation. On the other hand, there are several disadvantages, such as an uncertainty of the lowest achievable dark current even with deep cooling, some problems with photometric accuracy, a lack of sCMOS heritage in space and the overall developing risks related to a new custom UV-optimised sCMOS. Because of these reasons, a more conservative approach to using mature CCD technology was selected.

The main advantages of the CCD detectors compared to MCP are high sensitivity in the NUV range, a large format and geometrical stability, high dynamical range and the possibility to obtain UV spectra with a high SNR from relatively bright targets without the risk of damaging the detector.

In the FUV range (120–200 nm), the sensitivity of the CCD is decreased due to the absence of suitable space-qualified CCD AR coatings operated in this range. Therefore, for weak sources, the CsI MCP operated in photon-counting mode still has better sensitivity than CCD in FUV.

The WUVS detector subsystem was designed based on the Institute of Astronomy RAS (INASAN) statement of work and produced by Teledyne e2v and RAL space companies, and the main components are shown in Figure 4. It consists of three channels (Figure 5), each optimised for a specific range of wavelengths. For design reasons, all three detectors should be identical, except for minor changes such as anti-reflection coating on the CCD, the selection of the readout area of the CCD and active output amplifiers. Three variants of a custom CCD272-64 sensor with different UV AR coatings, each optimised for a specific WUVS channel, were designed by the team.

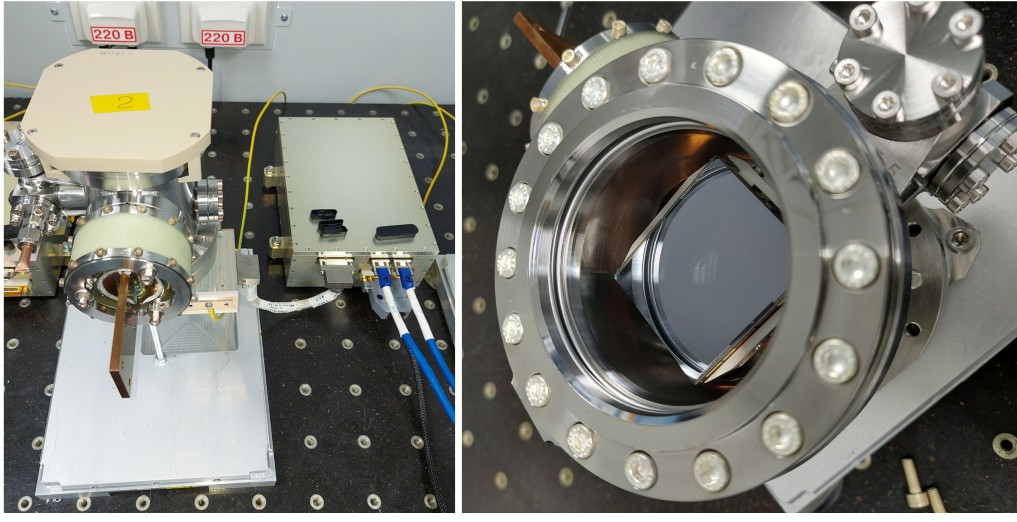

**Figure 4.** The engineering qualification model (EQM) of the enclosure with interconnection module (ICM) and camera electronics box (CEB).

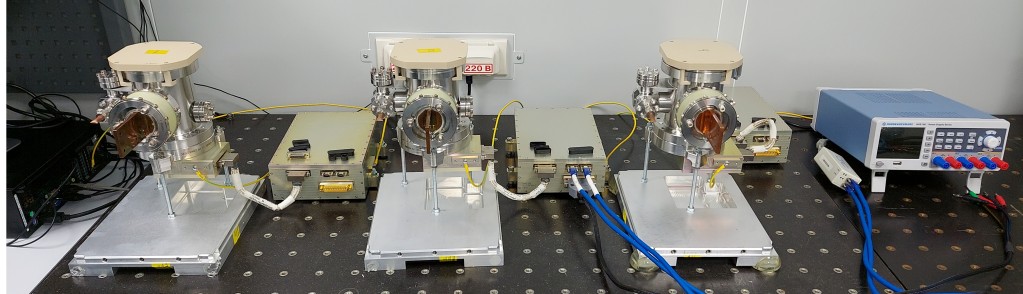

**Figure 5.** Three WUVS detector assemblies ready for installation.

The CCD quantum efficiency (QE) in NUV and especially in FUV is very sensitive to the CCD surface molecular contamination. The WUVS optical assembly and T-170M

telescope scientific instrument compartment are not clear enough to allow for the operation of an open-face cooled CCD in the UV range. To prevent contamination of the CCD and maintain the CCD operating temperature at $-100\,°C$, a custom vacuum enclosure with an input window made of $MgF_2$, identical for all WUVS channels, was designed. The drawbacks of this design approach are the light losses on the window and additional thermal load on the CCD cooling system. The $MgF_2$ sample demonstrated a transmittance of 67% at 120 nm and 94% at 300 nm wavelengths.

The CCD enclosures and CCD drive electronics are located inside the WUVS optical assembly because the distance between the CCD and electronics should be minimised.

To prevent the WUVS optical bench being heated by CCD drive electronics, electronic boxes are mounted on the thermal inserts and each electronic box has a heat pipe to evacuate its heat (about 10 W) to the WUVS wall that acts as a large common radiator. The WUVS optical scheme will be aligned at a $+20\,°C$ temperature and the WUVS optical assembly should be stabilised at this temperature after the launch; therefore, the CCD enclosure mounting feet will also be at $+20\,°C$.

Mechanical shutters are located on the top of the WUVS optical assembly slightly above the spectrographs' entrance slits. Their main goals are to block the light during the CCD readout procedure and to obtain CCD bias and dark calibration frames [8].

The WUVS spectral resolution element (resel) has a size of about 70 μm. It was a tradeoff in the WUVS custom CCD pixel size between 12 μm and 24 μm. For a small CCD pixel, the WUVS resel has a size of six pixels and, for a large pixel, three pixels. The choice is not so obvious, especially if the CCD operates in a space environment. The main differences with ground-based spectrographs are the presence of cosmic rays on each frame, quick degradation of CCD charge transfer efficiency (CTE) with time, especially for low-level signals, and quick general degradation of the CCD cosmetics.

In the case of having a large pixel with a notch channel, the low-level signal CTE for a combined signal within the resel will potentially be higher and the CCD readout noise contribution will potentially be lower.

With a small pixel, these two factors act in opposite (negative) directions. But small pixels also have advantages in operating the CCD in a space environment. Firstly, it is possible to reject cosmic rays more gently, which is a large problem for WUVS long-exposure images with a close to zero background. Therefore, increasing the subexposure time slightly might be considered, which leads to a decrease in the amount of CCD readouts within the long exposure, and it will reduce the CCD readout noise contribution. The other benefit of a small pixel is to have a higher tolerance for the appearance of bad columns and hot pixels after 5–10 years of operation in space. Indeed, if the resel has a size of $6 \times 6$ pixels, the presence of several remarkably defected pixels may be tolerated. And, finally, a small pixel always provides the best spectra sampling.

WUVS science and technical teams have decided to choose the small pixel CCD option. It will maximize the scientific performance during the first few years of the WSO-UV mission when the CCD will not be seriously degraded by radiation. During the second half of the WUVS life in space with a degraded CCD, it provides the best WUVS performance for observing medium and bright sources.

There is a tradeoff between using a classical single-end CCD video signal output and pseudo-differential video output. When using both real and dummy CCD outputs, there is better common-mode noise rejection and reduced susceptibility to electromagnetic interference and interference from the CCD clocking. To minimise the WSO-UV mission risk, pseudo-differential video output was selected.

Each WUVS detector has a small tungsten lamp to illuminate the CCD in optical wavelength. Before each sub-exposure, this system provides a preflash of the CCD at a level of about $10\ e^-$/pixel to fill the traps to improve charge transfer efficiency. The second purpose of the lamps is to obtain more or less uniform flat field images at a level of a few thousand electrons per pixel. This helps for the quick evaluation of the CCD pixel health as it identifies new bad or hot pixels because of the quick CCD degradation in space. Due to

the illumination system operating in an optical wavelength, these flat fields cannot be used to calibrate scientific UV spectra.

CCD detectors are very sensitive to optical wavelength radiation; therefore, all internal elements of the WUVS have a black coating optimised for visible light suppression. Some residual light from the Sun may enter the telescope's scientific instrument compartment through its side walls via small holes and gaskets. The WUVS optical assembly was designed to be light-tight, so, together with the telescope, there is double-layer protection for the WUVS detectors from Sun radiation.

Because of the compact and crowd optical design, the WUVS has a limited amount of space to install internal baffles and diaphragms. Each detector has a baffle in front of the CCD to reject out-of-band spectral orders; nevertheless, VUVES and UVES detectors remain partially open for scattered light inside the WUVS. The LSS channel has an additional baffle to avoid direct illumination of the detector from the grating.

## 3. The WUVS CCD

The WUVS CCD272-64 is a derivation of CCD273 used in the ESA's EUCLID mission; therefore, we can classify the WUVS CCD as a semi-custom device [9,10]. The main modifications are the higher gain of the output amplifiers to achieve better noise performance and back surface optimisation to operate in the UV range.

WUVS semi-custom CCD272-64 is a back-thinned back-illuminated two-phase device pixel array, the format is 4096 columns by 3112 rows and the pixel size is 12 μm square. Depending on the WUVS scientific observation program, pixels can be combined in $2 \times 2$ groups to give an effective 24 μm pixel size for minimisation of the readout noise within the resolution element of the spectrum and/or to decrease the readout time.

CCD272-64 has two serial registers with two output amplifiers each, but the drive electronics only have two analog circuits to read the CCD. Because of this restriction, the CCD readout schemes are different for different channels in order to achieve the best performance.

The LSS version has been designed to read only a half of the chip using one serial register with its two (left and right) output amplifiers. The VUVES and UVES CCDs have been designed to read the whole chip using two serial registers, but each register reading uses only one amplifier.

One of the key scientific design drivers for the WUVS CCD is to optimise its performance in order to operate at very-low-level signals down to a few electrons per pixel with a very long integration time of up to 10 h. For this purpose, the CCD operation temperature has been chosen at $-100\,^\circ$C. At this temperature, an increase of 1 $^\circ$C in CCD temperature will result in about a 1.5 e$^-$/pixel/hour increase in dark current. The CCD uses a low-voltage process to minimise power consumption both on the device and in the drive electronics.

To implement cosmic ray rejection algorithms, the WUVS very long exposure will be split into a series of short sub-exposures of about 10 min each. This naturally leads to an increase in the readout noise.

One of the challenges of the WUVS CCD is to achieve a detector lifespan of up to 10 years while operating at the geosynchronous orbit. The radiation degradation of the CCD leads to the increasing of a dark current, number of cosmetic defects and, the most critical parameter for the WUVS spectrograph, the degradation of charge transfer efficiency (CTE) for low-level signals on a zero background. To prolong the long-term stability of the WUVS CCD detector performance in a radiation environment, several techniques and design features are used:

- Shielding of the CCD by enclosure and other mechanical elements of the WUVS and spacecraft;
- Using the Teledyen e2v "radiation hard" process;
- Putting the CCD parallel transfer direction perpendicular to the dispersion of each WUVS channel. This design choice helps to minimise degradation of the spectral

resolution due to CTE, as parallel transfer CTE is higher than serial transfer CTE, but the consequence of this choice is a higher degradation of spectral line intensity because of parallel transfer CTE, e.g., smearing and charge losses;

- Putting the LSS channel FUV spectrum directly next to the serial register to minimise the number of parallel transfers, and the NUV spectrum next to the FUV, because, typically, the FUV spectrum is weaker than the NUV spectrum;
- Using a split frame transfer CCD readout mode to minimise the number of parallel transfers for VUVES and UVES channels;
- Using an optical preflash system to illuminate the CCD at a level of about 10 e$^-$ for partially filling charge traps;
- Heating the CCD up to about +20 °C periodically for annealing.

For the WUVS science program, the best possible readout noise is more important than the full well capacity, so the conversion factor of the CCD output amplifiers increased to 7 µV/e$^-$, and, accordingly, the maximum signal in a pixel decreased to 30,000 e$^-$. Such a full well capacity would not limit the ability to obtain high SNR spectra, because one resolution element has a size range from $3 \times 3$ to $6 \times 6$ pixels, and all exposures will be split into sub-exposures. The CCD gain is 0.8 e$^-$/ADU in a 16 bits format. To improve common mode noise suppression, a differential output architecture of the CCD output circuits is used.

One of the problems of the CCD operation in space is general degradation of the CTE. Additionally, the low-level signal CTE with zero background degradation needs to be considered. For the WUVS, the second aspect is the most important because typical spectra have a very low intensity while the spaces between echelle spectral lines are completely dark.

Low-level signal CTE with zero background degradation occurs because of the radiation degradation of the silicon, which causes many small traps to appear. If the traps are empty, they can temporarily catch several electrons during integration or CCD readout; therefore, some low-level signal details on the image may smear or even completely disappear. After a few years of operation in space, the low-level signal CTE with zero background may be more than an order of magnitude lower than CTE for a high-level signal with non-zero background.

The CCD272-64 pixel structure forms a potential well within the pixel that acts as a "notch" channel. While the amount of collecting electrons in the pixel is small, they are concentrated in the center of the pixel and, therefore, the probability to be captured by traps is reduced. During an on-ground validation campaign, WUVS CCD272-64 was tested before and after irradiation to measure both standard CTE and low-level signal CTE with zero background.

One of the main challenges for the WUVS CCD is to provide the best possible quantum efficiency over a very challenging spectral range from 120 nm to 320 nm. UV radiation is the most difficult to detect with CCD detectors because the absorption depth in silicon is the smallest at these wavelengths. The CCD must be thinned to minimise any dead layer at the back surface and a different kind of treatment of the back surface may be implemented to further improve the quantum efficiency. Teledyne e2v has chosen to use a laser annealing treatment of the WUVS CCD. The drawback of this technique is the presence of a fixed pattern variation of QE in the UV range.

A novel process is used whereby the CCD has an anti-reflection HfO coating of different thicknesses over a part of the image area to match the operation wavelength at each point of the detector with the spectrograph's dispersion. The coating should be removed for the shortest wavelengths (115–180 nm), where the presence of the coating would degrade the quantum efficiency. Therefore, the VUVS channel's variant of the CCD has no coating at all, the UVES channel's CCD is fully coated with gradient AR coating and the LSS channel's CCD has two sections: an uncoated area for the FUV spectrum and a gradient AR coating area for the NUV spectrum.

## 4. The Hermetic CCD Enclosure

Due to the CCD cold surface being very sensitive to contamination, a custom cryostat (enclosure) was designed and manufactured to maintain a very clean environment for the CCD. The enclosure was designed according to ultra-high vacuum standards. The purposes of the enclosure are to:

- Protect the CCD surface from any kind of contamination;
- Provide an additional radiation shielding for the CCD;
- Provide a UV transparent window for illuminating the CCD;
- Provide a thermal path for cooling the CCD.

One of the challenges of the enclosure is to guarantee the absence of any condensation on the cold-sensitive CCD surface, which may impact the QE, for 9 years of operation. According to our calculation, the partial pressure of water within the enclosure should be maintained below $1.6 \times 10^{-5}$ mbar at $-100\,^{\circ}$C. Before the final assembling of the WUVS flight model, all enclosures will pass through a pump and bake procedure again and be filled up with ultra clean Ar.

The enclosure input window is maintained at +22 $^{\circ}$C by utilising an additional dedicated heater in order to minimise contamination on the window side facing towards the WUVS optical assembly. A spacecraft battery directly powers the CCD window heater to ensure the protection of the window against contamination. This is essential when the WUVS remains unpowered during the launch and commissioning phases and if the spacecraft enters a safe mode operation.

To prolong the detectors' lifespan in a radiation environment, the annealing procedure will be carried out monthly. For this purpose, a 500 W heater is installed on the external radiator for temporal deactivation of the CCD passive cooling system.

## 5. CCD Drive Electronics

The camera electronics box (CEB) was designed according to the INASAN statement of work and built by RAL Space, STFC (UK) based on its heritage of the production of CCD camera electronics for several projects, such as STEREO SCIP/HI, SDO AIA/HMI, GOES-R SUVI. The CEB is connected to the WUVS by a redundant SpaceWire link that is used for CCD image transmission, telemetry and the commanding of the CEB. The WUVS also provides redundant 27 V power for the CEB.

Three identical CEBs are located inside the optical-mechanical unit of the WUVS close to the dedicated CCD enclosures to minimise the analog cable lengths. The CEB power consumption is 11 W per channel and the heat produced by each CEB is evacuated by a dedicated heat pipe. The CEB box dimensions are $150 \times 220 \times 65$ mm, the weight is 3.2 kg and the wall thinness is 6 mm to improve protection against radiation (Figure 6).

The CEB houses three camera electronic cards: a power supply card, bridge card and CCD camera card. The CCD camera card provides the majority of the functionality within the CEB system, including the SpaceWire interface, video digitisation, CCD bias voltage generation and CCD clock driving.

A key technology implemented in the CEB is a digital correlated double sampling. It enables several readout speeds of 50 kHz, 100 kHz and 500 kHz, each in normal and binned pixel readout modes, with each mode optimised to have the lowest possible readout noise. The CEB 14-bit ADCs continuously sample the entire CCD video signal at a rate of 25 MHz, including settling and clamp periods. At a pixel frequency of 50 kHz, the CEB obtains about 160 valid samples to measure each pixel's signal. The average pixel value is calculated with high precision and then rounded to 16 bits to avoid an increase in system noise due to the quantisation error of 14 bits format data.

An additional interconnect module was also designed, which is mounted directly on the CCD video signal connector on the bottom of the enclosure. It provides the CCD video signal amplification to reduce the system noise and additional filtration of the CCD voltages. The interconnect module is connected to the CEB by a short electrical harness.

Two types of harnesses (VUVES/UVES and LSS) were developed in order to keep them as short as possible within the WUVS mechanical constraints.

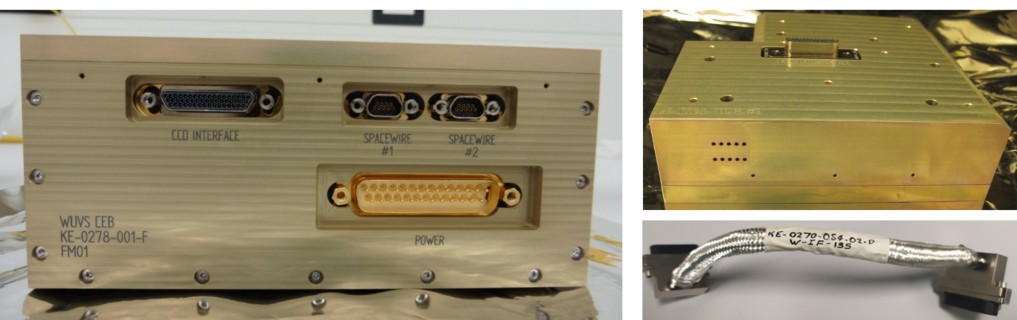

**Figure 6.** Flight models of the camera electronic box (CEB), interconnection module (ICM) and analog video cable.

## 6. Qualification Campaign Main Results

The assessment of quantum efficiency in FUV and NUV ranges for modern CCD and CMOS is still limited due to the complexity of the necessary equipment, especially the source of monochromatic, well-regulated and uniform UV radiation.

WUVS CCD reflectivity was measured to predict QE in the UV range using Teledyne e2v's own technique. In 2019, a direct QE measurement of the LSS channel engineering qualification model CCD was carried out on the metrological station "Kosmos" at the Budker Institute of Nuclear Physics. As a source of UV light, a synchrotron radiation from the VEPP-4M storage ring was used with a MgF$_2$ filter to cut off high-energy radiation and with the UV monochromator to select the working wavelength. All the measurements were performed at the CCD nominal working temperature of −100 °C. In addition to the QE measurement, the efficiency of the CCD enclosure cooling system was checked.

The LSS device has both an uncoated region and a gradient AR-coated region (Figure 7). In accordance with the WUVS spectrograph dispersion, the AR coating thickness is optimised for 180 nm on the left side of the CCD and for 310 nm on the right side.

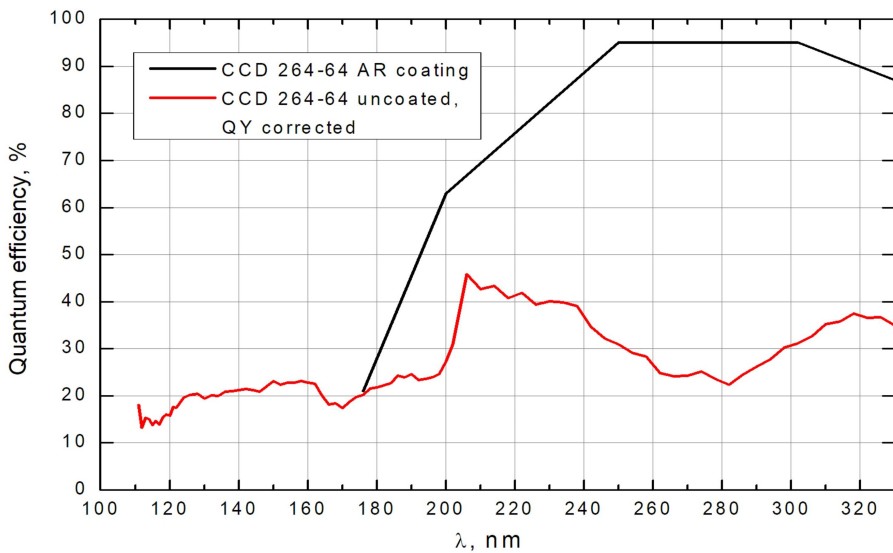

**Figure 7.** EQM LSS enclosure quantum efficiency measurement results for the uncoated and gradient AR-coated regions of the CCD.

The QE was measured in the 111–320 nm range for the uncoated region and at 10 points of different thicknesses for the AR-coated region. The measurement results show a good compliance with our prediction based on a reflectivity method. Our measurements show

up to a four times QE increase due to the AR coating, which is a great improvement in the CCD responsivity in UV. For future FUV missions, it is a great opportunity to develop new kinds of space-qualified anti-reflection coatings to operate below 180 nm.

Enclosure cooling system verification results confirm that the temperature difference between the CCD and the "cold finger" is 5 °C.

During the test of the FM CEB, a readout noise of about 2.6 e$^-$ RMS at 50 kHz and about 3 e$^-$ RMS at 100 kHz was measured. The linearity is better than 0.2% and the cross-talk between the channels is less than 29 ppm. The CEB linearity is about 0.1% within the full CCD input voltage range. The CEB electronics contribution to the crosstalk between two CCD readout channels is about 40 ppm, and was measured using a CCD signal emulator.

## 7. Summary and the Current Status

The WUVS detector has a custom design to maximise the scientific efficiency of the WSO-UV mission. Several methods for extending the service life of the CCD detectors in a space radiation environment were implemented.

The WUVS spectrograph CCD detector system has been developed based on the Teledyne e2v and RAL space expertise, considering specific requirements of the WSO-UV mission to operate with very low signals for a long exposure time.

The critical design review (CDR) of the camera electronics box and enclosure were passed in 2016 and 2019, respectively, while, between 2020 and 2021, the WUVS detector subsystem successfully passed the qualification campaign and the key parameters were verified. In addition to the Teledyne e2v factory validation and verification campaign, the quantum efficiency of the LSS EQM CCD and the cooling system efficiency were measured. In 2019, four CEB FM (camera electronics box flight model) units were successfully delivered to Russia and passed incoming inspection.

According to the qualification campaign results, the current WUVS detector design is mature enough to assemble the WUVS flight model.

**Author Contributions:** Investigation, writing—original draft preparation, A.S.; writing—review and editing, supervision, M.S. All authors have read and agreed to the published version of the manuscript.

**Funding:** This research received no external funding.

**Institutional Review Board Statement:** INASAN expertise on the possibility of publication N98/23 from 9 August 2023.

**Data Availability Statement:** Data available on request due to restrictions.

**Acknowledgments:** We express our gratitude to the Teledyne e2v and RAL Space companies for the production of the custom CCD272-64, custom enclosures and low readout noise electronics for the WSO-UV space project, and to the Lebedev Physical Institute for their continued project support and for organising the verification campaign. Without hard work and significant contributions from the whole WUVS project team the development of the WUVS detectors would have not been possible, in addition special thanks goes toward the BINP team for the provision of the facilities for characterisation of the CCD in UV.

**Conflicts of Interest:** The authors declare no conflict of interest.

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
