# Peer review of "Spektr–UF Mission Spectrograph Space Qualified CCD Detector Subsystem"

_photonics, doi:10.3390/photonics10091032_

Round 1

Reviewer 1 Report

The paper calls: "SPEKTR–UF MISSION SPECTROGRAPH SPACE QUALIFIED CCD DETECTOR SUBSYSTEM" and concerned of design of space spectrograph operating in UV spectra band. The main advantage of article is real experimental setup as topic of presentation. The design of setup is supported by references list.

However article have some optional disadvantages:

1. Optical scheme of ray tracing is absence.

2. Information of electrical power supply and consumption of setup is missing.

Author Response

Dear Referee,

Thank you for revision of the article and for useful comments to improve it.

What we did:

  1. Added: Optical schemes of WUVS spectrograph (Fig 2,3).
  2. Added: CCD drive electronics power consumption (section: CCD drive electronics).

Reviewer 2 Report

Article with proper: title, organization, division in sections is proposed. The subject of the study is suitable for journal Photonics. The abstract is concise, introduction is informative and sufficient. The essence of the work is very well presented. The working parameters, new achievements, advantages, and main strengths of the elaborate World Ultraviolet Space detector system are outlined. The quality of the included images and figures is very good. The illustrated results in figure 4 support the claims and final summary.

The text could be improved to be more readable for the audience, particularly for students, young researchers, and those from another scientific fields. There are several abbreviations that are not introduced when the first time appear in the text:  

Figure caption of figure 2: “The engineering qualification model (EQM) of the Enclosure with ICM and CEB.” Abbreviations ICM and CEB should be given in text as engineering qualification model (EQM). Camera Electronics Box (CEB) appears for the first time in 286 line.

Line 80: “sCMOS”

Line 89: “SNR”

Line 128: Charge transfer efficiency should appear after “CTE”.

Line 153: Is it “lump”?

Line 208: “to put the CCD parallel transfer direction perpendicular to the dispersion of each WUVS channel, as parallel transfer CTE is higher than serial transfer CTE;“ The sentence should be clarified to be understandable.

Line 250: “QE”, “quantum efficiency” is given later in line 257.

Line 331: (Figure 4) should appear earlier in the text, for example in line 327.

Line 350: “Camera Electronics Box” instead “CEB” sounds better.

Author Response

Dear Referee,

Thank you for revision of the article and for useful comments to improve it.

What we did:

Abstract: EQM -> engineering qualification model

figure 2, ICM -> interconnection module (ICM), CEB -> Camera Electronics Box (CEB

sCMOS  -> scientific Complementary metal–oxide–semiconductor (sCMOS)

SNR   was described at the second paragraph of section 2

CTE  -> charge transfer efficiency (CTE)

lump  -> lamp

“to put the CCD parallel transfer direction

perpendicular to the dispersion of each WUVS channel, as

parallel transfer CTE is higher than serial transfer CTE;“  ->

to put the CCD parallel transfer direction perpendicular to the dispersion of each WUVS channel, this design choice helps to minimise degradation of the spectral resolution due to CTE, as parallel transfer CTE is higher than serial transfer CTE, but the consequence of this choice is higher degradation of spectral line intensity because of parallel transfer CTE, e.g. smearing and charge losses;

quantum efficiency -> quantum efficiency (QE)

Line 331: (Figure 4) should appear earlier in the text, for example in line 327. - done

Line 350: “Camera Electronics Box” instead “CEB” sounds better. - done

Additional small mistake found:

We had measured WUVS CCD reflectivity to predict QE in the UV range using 319

their own technique. -> WUVS CCD reflectivity had measured to predict QE in the UV range

using Teledyne e2v own technique.

Reviewer 3 Report

The manuscript "SPEKTR–UF MISSION SPECTROGRAPH SPACE QUALIFIED
CCD DETECTOR SUBSYSTEM" by Andrey Shugarov and Mikhail Sachkov,
describes Ultraviolet spectrograph detector design drivers, methods for extending the service life of the CCD sensors working with low signals in a space radiation environment, the key calculated parameters and results of the EQM (engineering qualification model) qualification campaign.All this for Spektr-UF (World Space Observatory) The paper is well written, informative and of interest for wider public. I recommend its publication in
Photonics.

Before publication, at the end of Introduction, a couple of sentences should be added to explain the aim and content of this work.

Misprints:

L91: due to and absence ?

L153: lump --> lamp

L251: presents --> present

The manuscript "SPEKTR–UF MISSION SPECTROGRAPH SPACE QUALIFIED
CCD DETECTOR SUBSYSTEM" by Andrey Shugarov and Mikhail Sachkov,
describes Ultraviolet spectrograph detector design drivers, methods for extending the service life of the CCD sensors working with low signals in a space radiation environment, the key calculated parameters and results of the EQM (engineering qualification model) qualification campaign.All this for Spektr-UF (World Space Observatory) The paper is well written, informative and of interest for wider public. I recommend its publication in
Photonics.

Before publication, at the end of Introduction, a couple of sentences should be added to explain the aim and content of this work.

Misprints:

L91: due to and absence ?

L153: lump --> lamp

L251: presents --> present

Author Response

Dear Referee,

Thank you for revision of the article and for useful comments to improve it.

What we did:

At the end of Introduction, we add 4 sentences to explain the aim and content of this work.

L91: due to and absence ?   -> due to the absence of

L153: lump --> lamp - done

L251: presents --> present     -> presence